# Public perceptions and support for introduced microbes to combat hospital-acquired infections and antimicrobial resistance

Christopher L. Cummings[1]*, Kristen D. Landreville[1], Jennifer Kuzma[1,2]

1 Genetic Engineering and Society Center, Raleigh, North Carolina, United States of America, 2 School of Public and International Affairs, North Carolina State University, Raleigh, North Carolina, United States of America

* christopherlcummings@gmail.com

## Abstract

Hospital-acquired infections and antimicrobial resistance (AMR) remain major global health threats, prompting interest in microbiome-based interventions that introduce beneficial microbes or genetic interventions to control pathogens and reduce AMR genes in hospital environments. Microbiome engineering, which can use advanced biotechnology, genetics, and microbial ecology principles to restructure microbial communities, is a rapidly growing field with applications in infection control. As researchers explore deploying beneficial microbes and other genetic interventions in clinical settings like hospital sinks, public perception becomes critical to responsible implementation. This study addresses how U.S. adults perceive microbiome evaluation, and education. Drawing on a nationally representative survey (N = 1,000), we conducted hierarchical ordinary least squares regression modeling to assess predictors of support across three domains: implementation of introduced microbiomes (IM), rigorous testing, and education for healthcare stakeholders. Results demonstrate that support for IM in hospital sinks is shaped less by demographic traits and more by emotional responses, trust in institutional efficacy, belief in intervention benefits, and a desire to learn about microbiome science. These findings advance previous knowledge by distinguishing cognitive, affective, and contextual predictors across distinct types of support. Contrary to expectations, prior familiarity and information-seeking were negatively associated with IM support, suggesting that some engagement or exposure to risk-framing may drive skepticism. Meanwhile, emotional reactions and perceived efficacy consistently predicted support for IM, testing, and education (i.e., across all dependent variables), underscoring the need to address affective and trust-based components of public engagement. This research contributes to an emerging empirical foundation for responsible microbiome innovation by grounding the analysis in the Responsible Research and Innovation (RRI) framework. With the technology still in early development, these insights provide critical guidance

**Data availability statement:** The data underlying this study cannot be shared publicly due to ethical and legal restrictions imposed by the North Carolina State University Institutional Review Board (IRB). Although the dataset is de-identified, it contains potentially sensitive information related to participants' health beliefs, religious practices, political orientations, and emotional responses to biotechnological interventions—data that, in combination, could pose a risk of deductive disclosure. The IRB determined that public release of the dataset, even in de-identified form, could not ensure adequate protection of participant privacy given the nature of the topics studied and the granularity of some variables. This determination aligns with established ethical standards in human subjects research that prioritize respect for persons, confidentiality, and the principle of minimizing harm. Access to the de-identified data may be requested on a case-by-case basis for qualified researchers who agree to strict confidentiality protocols and data use agreements. Requests should be directed to: North Carolina State University Institutional Review Board Email: irb-director@ncsu.edu Phone: +1 (919) 515-4514 Website: https://research.ncsu.edu/compliance/irb/.

**Funding:** The Center for Precision Microbiome Engineering (PreMiEr) is a National Science Foundation (NSF)-funded Engineering Research Center (ERC) (Award # 2133504). The funders had no role in study design, data collection and analysis, decision to publish, or preparation of the manuscript.

**Competing interests:** The authors have declared that no competing interests exist.

for biotechnology developers, policymakers, and hospital leaders seeking to align microbiome engineering with societal values through transparent communication, rigorous oversight, and inclusive education.

## Introduction

Hospital-acquired infections (HAIs) pose a significant safety risk to both healthcare providers and patients, with particularly severe consequences for immunocompromised individuals [1,2]. These infections can lead to substantial morbidity and mortality, prolong hospital stays, and complicate patient recovery. The financial burden associated with HAIs is immense, as they contribute to increased healthcare costs, reputational damage to hospitals, and additional demands for reimbursement [3,4]. In the United States alone, it is estimated that HAIs cause or contribute to approximately 99,000 deaths annually, underscoring the urgent need for effective infection control strategies [5]. As a result, infection prevention has become a top national healthcare priority in many developed nations, prompting calls for evidence-based interventions and systemic changes to enhance infection control measures [6].

As HAIs continue to pose significant public health risks, microbiome engineering has emerged as a promising, innovative strategy for infection control. Microbiome engineering stands at the intersection of microbiology, genomics, and engineering, providing novel opportunities to modify microbial communities across different environments [7]. This emerging field focuses on strategically altering microbial composition and function to promote health benefits, such as reducing disease risks and enhancing environmental quality [8,9]. The built environment, which includes the spaces where people live, work, receive medical care, and socialize, has become a key area of research for microbiome interventions aimed at improving public health [10]. By leveraging microbiome engineering, scientists can potentially restructure microbial ecosystems within these environments, transforming them into actively health-promoting spaces that contribute to human well-being. One promising area under development is the potential use of introduced microbes or microbiomes in hospital settings to combat antimicrobial resistance (AMR) by limiting the spread of resistant pathogens and promoting healthy microbial communities. This study focuses specifically on hospital sinks as the target site for introduced microbiomes due to their well-documented role as persistent reservoirs for multidrug-resistant organisms such as *Pseudomonas aeruginosa, Klebsiella pneumoniae*, and *Acinetobacter baumannii*. These pathogens frequently colonize sink drains and plumbing systems, where they can form biofilms, exchange resistance genes, and contribute to repeated contamination events within clinical environments. Sinks represent one of the few built-environment surfaces in hospitals currently being explored for targeted microbial interventions, including probiotic and synthetic consortia-based approaches. As such, they offer a realistic and policy-relevant use case for gauging public support for microbiome engineering in infection control. Novel technologies, such as using Clustered Regularly Interspaced Short Palindromic Repeats (CRISPR)-based gene-editing either delivered by microbes or particles, are also being tested for use to

degrade antimicrobial resistant genes that spread to the harmful microbes that plague hospital sinks and contribute to the hundreds of thousands of HAIs each year [11].

Studies have explored engineered bacterial consortia and probiotic interventions as potential strategies for suppressing multi-drug resistant (MDR) bacteria in healthcare environments [12]. For example, precision-engineered waste lignocellulosic biochars have been shown to sequester multidrug-resistant clinical isolates in model wastewater systems (removing up to 94% of Pseudomonas aeruginosa and 85% of Staphylococcus aureus) — though their efficacy in clinical or hospital environments has not yet been evaluated [13]. Although recurrent infections associated with Pseudomonas and biofilm formation on medical devices remain a serious clinical challenge [14], emerging studies suggest that introduced non-pathogenic microbes may help outcompete pathogenic strains in certain contexts. For example, experimental trials using engineered bacterial consortia have demonstrated success in displacing Klebsiella pneumoniae and other multidrug-resistant pathogens in controlled environments [15]. Such microbial-based approaches align with antimicrobial stewardship efforts, emphasizing the importance of proactive and preventative strategies to mitigate AMR risks in hospitals. As research advances, incorporating synthetic biology, bacteriophage therapies, and microbiome engineering into infection control policies may provide sustainable solutions for combating AMR in hospital settings.

In the United States, where advancements in biotechnology and microbiome research are at the forefront, public opinion plays a critical role in shaping the trajectory of these emerging technologies [16]. The potential application of genetically engineered microbes to hospital environments represents an exciting yet complex frontier, requiring a balance between scientific advancement, patient, hospital-worker, and overall societal acceptance. Before large-scale deployment, understanding public attitudes, trust in microbiome interventions and the groups overseeing them, and perceptions of risk and benefit are essential to ensure successful integration into healthcare settings [17]. Despite their potential benefits, microbiome-based interventions raise ethical, regulatory, and societal considerations, particularly regarding equitable access, informed consent, oversight, and unintended consequences [18]. Public trust in these technologies is heavily influenced by scientific communication, trust in institutions to effectively use technologies while mitigating risks, and perceived trade-offs between risks and benefits of interventions (Cummings, Rosenthal, & Kong 2020, [19,20]. Previous research has indicated that early public engagement in the development process of emerging biotechnologies is crucial for ensuring widespread acceptance and responsible innovation (Cummings and Peters, 2022) [21]. Integrating public perspectives into microbiome engineering research not only fosters transparency but also ensures alignment with societal values and ethical standards.

To address this, researchers affiliated with a National Science Foundation-funded Engineering Research Center on Precision Microbiome Engineering (Award # 2133504) are actively incorporating a Responsible Research and Innovation (RRI) approach. Through a nationwide representative survey of US adults, this initiative seeks to assess public attitudes, concerns, and expectations regarding microbiome engineering for infection control in hospitals [16]. By actively engaging stakeholders and the broader public, this research seeks to facilitate dialogue between scientific advancements and societal perspectives, fostering informed discussions that can guide responsible innovation and implementation.

To ensure that microbiome-based interventions align with societal values and ethical considerations, it is critical to examine how the public perceives both the risks and benefits of introducing microbes in hospital settings. Public trust in emerging biotechnologies is shaped by multiple factors, including the credibility of the institutions overseeing their development, transparency in risk communication, and perceived trade-offs between potential health benefits and unintended consequences [16,17]. Effective governance and regulatory frameworks must therefore incorporate public concerns and expectations to support the ethical deployment of microbiome engineering in healthcare environments. Furthermore, given the historical challenges in public acceptance of genetically modified organisms (GMOs) and synthetic biology applications, proactive engagement strategies are necessary to address misconceptions, promote informed decision-making, and ensure that innovations are developed in a socially responsible manner [22,23]. This study is therefore guided by the following research question: *"How do U.S. adults perceive the introduction of microbiomes in hospital environments to support the growth of good microbes and control the growth of harmful ones?"*

## Literature review

### Responsible research and innovation (RRI)

Responsible Research and Innovation (RRI) is an emerging governance framework developed in the early 2010s by scholars in science and technology studies, primarily in Europe, as a response to the growing recognition that traditional approaches to risk governance inadequately address the broader social and ethical dimensions of emerging technologies. Rather than focusing narrowly on technical risk assessment, cost-benefit analysis, or regulatory oversight, RRI emphasizes the importance of aligning scientific research and innovation with societal values, democratic deliberation, and ethical reflection [24]. The framework builds on the legacies of earlier efforts, such as ELSI (ethical, legal, and social implications) in the U.S. and ELSA (ethical, legal, and social aspects) in Europe, but extends these by emphasizing proactive and participatory engagement throughout the research and development process [25].

Stilgoe et al. [24] argue that public controversies surrounding science and technology often stem not solely from perceived risks, but from deeper concerns about the goals, values, and purposes of innovation itself. As such, RRI promotes what Owen, Macnaghten, and Stilgoe [26] describe as a shift "upstream" in governance—encouraging reflection on the motivations, design choices, and social contexts of research before technological trajectories become fixed. The RRI framework is structured around four foundational principles: anticipation, inclusion, reflexivity, and responsiveness [24]. Anticipation involves exploring possible future outcomes and impacts of research, including unintended consequences, in order to inform more socially robust innovation. Inclusion calls for the integration of diverse perspectives—including those of lay publics, marginalized groups, and non-experts—into research processes, moving beyond technocratic decision-making. Reflexivity encourages researchers to critically examine their own assumptions, commitments, and epistemological frames, as well as the limits of their knowledge. Responsiveness refers to the capacity of research systems and institutions to adapt in light of new knowledge, public concerns, or emerging social priorities. Although RRI has been formally adopted in European science policy, such as through Horizon 2020 funding mandates, its uptake in the United States has been far more limited, with few institutionalized pathways for integrating RRI into national research funding or innovation governance [27]. Nevertheless, RRI principles offer a valuable conceptual and practical guide for developers of novel biotechnologies, including microbiome engineering, particularly in building public trust and legitimacy.

As microbiome engineering and introduced microbiomes in built environments are in very early stages of development, an assessment of public attitudes can help to inform the future trajectory of the technologies. In this study, the RRI framework informs both the conceptual orientation and the empirical structure of our analysis. The regression models incorporate variables aligned with RRI's core principles: inclusion (e.g., demographic diversity), reflexivity (e.g., perceived knowledge and interest in learning), anticipation (e.g., emotional reactions and perceived consequences), and responsiveness (e.g., trust in hospital governance and health agencies). By embedding these dimensions into our model design, we aim to assess how public support for microbiome interventions reflects the broader values and concerns emphasized in RRI. This approach enables a more nuanced understanding of how publics engage with emerging microbial technologies and provides empirical insights into how responsible innovation can be fostered in healthcare contexts.

### Public perceptions of hospital acquired infections

While, to date, the authors are the first to systematically evaluate public perceptions of introduced microbes for combatting AMR in hospital settings, related studies have evaluated public perceptions to HAIs as well as introduced microbes for intended beneficial purposes. Globally, public perceptions of HAIs are demonstrated to be critical in shaping hospital infection control efforts, influencing patient behavior, and informing healthcare policies. Misconceptions and limited awareness about HAIs persist among patients and the general public, often leading to a false sense of security or blame misattribution toward healthcare professionals alone [28]. For instance, a survey conducted in a large teaching hospital in Rome found that patients underestimated their personal risk of acquiring a HAI, and many perceived infection

prevention as the sole responsibility of hospital staff rather than a shared effort [28]. This aligns with related findings from Nepal, where increased patient trust in hospital infection control measures significantly influenced their willingness to participate in preventive actions, such as hand hygiene or questioning antibiotic prescriptions [29]. In Latin America, a study by Fabre et al. [30] revealed that cultural and socioeconomic factors heavily impact public attitudes toward infection prevention programs, with many communities showing hesitation or noncompliance due to misconceptions about the necessity of strict hygiene protocols. Similarly, research from German hospitals highlights the growing role of digital surveillance systems in infection control, yet public perceptions remain divided, with some viewing digital monitoring as a necessary tool for HAI prevention while others fear privacy risks and data misuse [31]. Another study from China emphasized the lasting impact of the COVID-19 pandemic on public attitudes toward hospital hygiene, where post-pandemic surveys indicated increased patient demand for stricter infection control policies in medical facilities [32]. Additionally, healthcare workers' perspectives on HAIs play a crucial role in shaping public perceptions, as demonstrated in a study from Saudi Arabia, where hand hygiene compliance among medical staff directly influenced public confidence in hospital infection control measures [19]. Research also indicates that high-profile hospital infection outbreaks significantly alter public perception, leading to heightened scrutiny of hospital hygiene practices and antimicrobial use [33]. The growing presence of AMR further exacerbates public concerns, as demonstrated in a study by Cirillo et al. [34], where the declining effectiveness of antibiotics in treating HAIs was associated with increased patient anxiety and distrust in healthcare institutions.

## Public perceptions of microbes and microbiome engineering

While microbiome engineering presents promising applications for healthcare, environmental management, and built environments, public acceptance remains divided due to varying levels of awareness, understanding, and risk perception. Research suggests that factors such as age, education level, trust in science, and exposure to nature significantly influence these perceptions ([16]; Robinson et al., 2021) [35]. So far there have been limited studies of U.S. public attitudes on microbiome engineering in the built environment. In the first survey of built microbiome engineering and public attitudes, attitudes toward microbes and microbiome engineering were discovered to be shaped by a complex interplay of scientific literacy, emotional responses, media influence, and demographic factors [16,36]. In this same survey [16], younger individuals, those with higher education levels, and those with greater trust in scientific institutions were more receptive to microbiome engineering in the built environment, whereas older adults and those with lower scientific trust exhibited skepticism and concern over unintended consequences. Hardwick et al. [23] further explored the societal and ethical implications of microbiome engineering in built environments, identifying concerns surrounding safety, equitable access, and regulatory oversight. For general microbiome engineering (not specific to the built environment), Parker and Kunjapur [37] emphasized the role of transparent research practices and regulatory frameworks in shaping public trust, suggesting that clear communication of potential benefits and safeguards plays a pivotal role in acceptance. However, Galli and Fasanelli [22] argue that media framing and public discourse often associate microbiome engineering with genetic modification, reinforcing ethical and safety-related anxieties.

Beyond microbiome engineering, the general perception of microbes also influences how the public views engineered microbial applications. A study by Kokkinias et al. (2024) [38] found that while microbes are frequently associated with disease, some participants acknowledged their beneficial ecological and health roles. However, confusion over microbial terminology, such as the distinction between bacteria and viruses, led to persistent negative associations, which influenced willingness to support microbiome research. Emotional responses, particularly disgust, also play a role in shaping public attitudes. Yeo et al. (2019) [39] demonstrated that disgust-inducing language heightened perceived risks of microbiome interventions in humans, particularly in the context of fecal microbiota transplants for treating Clostridium difficile infections. While microbiome-based therapies show clear medical benefits, emotional barriers may limit public acceptance, demonstrating potential need for more careful framing and communication strategies.

Perceptions among students also highlight a strong association between microbes and risk. For instance, Karadon et al. (2010) [40] reported that over 50% of school-aged children described microorganisms using terms like "dirt," "pollutant," or "harmful", with limited recognition of beneficial microbial roles. Although college students demonstrated improved microbial knowledge after completing undergraduate microbiology coursework, this did not always translate into behavior changes, such as increased vaccine uptake or personal infection control measures (Jones et al., 2012) [41]. These findings suggest that factual knowledge alone is insufficient to shift behaviors or reduce perceived risks, highlighting the need for targeted educational engagements that address both cognitive and emotional barriers. Large-scale public surveys further illustrate the knowledge gap surrounding microbes. A kiosk-based study at the American Museum of Natural History in New York City collected responses from over 22,000 visitors across 172 countries, revealing that only 50% of respondents correctly identified penicillin as an antibiotic, while fewer than half viewed microbes as beneficial (Zichello et al., 2021) [42]. The authors argue that low microbial literacy may hinder the acceptance of public health initiatives that rely on microbial applications, a concern echoed by Thaler et al. (2019) [43]. Additionally, Shan et al. (2019) [44] highlights the importance of clear and responsible scientific communication, warning that overstating the significance of microbiome research findings may erode public trust if perceived as exaggerated or misleading.

These studies collectively underscore the complexity of public perceptions regarding microbes and microbiome-based interventions. While increased scientific literacy can enhance understanding, knowledge alone does not necessarily translate into behavioral shifts or greater acceptance of microbial applications in healthcare settings. Emotional responses, prior experiences, and trust in institutions play crucial roles in shaping public attitudes, particularly when it comes to novel biotechnologies such as microbiome engineering. Given the potential for misinformation and uncertainty surrounding these interventions, it is essential to assess how the broader public perceives the introduction of beneficial microbes in hospital environments. Understanding these attitudes will provide valuable insights into the factors that influence support, skepticism, or opposition, thereby informing strategies for responsible implementation and public engagement.

## Educational support for healthcare staff and patients in microbiome-based interventions

The successful implementation of introduced microbiomes (IM) in hospital settings depends not only on technological innovation but also on the preparedness and awareness of those who interact most with healthcare environments, namely, staff and patients. Evidence demonstrates that many healthcare workers lack a full understanding of the hospital microbiome, despite being frontline agents in infection control [45]. In particular, the perception of all microbes as harmful persists even among trained staff, reinforcing sterile paradigms and undermining the acceptance of beneficial microbial interventions. This misinformation gap is exacerbated by patients' unfamiliarity with microbiome-related concepts, contributing to distrust in emerging microbial technologies. Tozzo, Delicati, and Caenazzo [46] underscore the need for educational strategies targeting both medical personnel and patients, advocating microbiome-based training as a core component of hospital infection control programs. Training initiatives must go beyond factual knowledge and include risk communication frameworks that address emotional and cultural factors influencing microbial perceptions [47].

Built environment studies further highlight the role of human-microbe interactions in hospital ecosystems. Bruno et al. [48] and Marotz et al. [49] demonstrated that hospital surfaces, staff movements, and patient behaviors all shape microbial landscapes. Without informed behavioral guidelines, these interactions may unintentionally foster pathogen persistence. Educational programming for healthcare workers, therefore, must incorporate the dynamics of built environment microbiomes and their relevance to infection control.

Training is also critical in managing concerns related to AMR. Sukhum et al. [50] found that antibiotic-resistant organisms persist even in newly constructed hospitals, pointing to the urgent need for staff education on microbial reservoirs and patient-staff transmission routes. Similarly, Dalton et al. [51] advocate for a "One Health" approach, which connects human, environmental, and microbial health and supports training initiatives that reflect this systemic complexity. The Hospital Microbiome Project further supports integrating microbial awareness into clinical education. Wilson et al. [52–55]

describes efforts to increase staff and patient understanding of how the indoor microbiome contributes to HAIs, noting the value of embedding such literacy into hospital innovation policies. Taken together, these studies reinforce the need for robust educational infrastructure to support the engineering of microbiomes in hospitals. By increasing microbial literacy, addressing emotional and cultural barriers, and enhancing risk communication, healthcare systems can better align microbial interventions with patient and staff values, ultimately fostering institutional trust and public support. To explore these dynamics, this study employs a nationally representative survey of U.S. adults, designed to assess perceptions, concerns, and expectations regarding the use of introduced microbes for infection control in hospital settings. The following section outlines the methodological approach used to gather and analyze public opinion on this emerging biotechnology.

## Method

### Sampling procedure

This study utilized a cross-sectional survey design to examine public attitudes toward microbiome engineering within the built environment. Data collection took place in December 2023, drawing on a nationally representative sample of 1,000 U.S. adults (18 years and older), sourced from YouGov's National Omnibus Panel. This study was reviewed and approved by the North Carolina State University Institutional Review Board (IRB Protocol #26517). All participants provided informed consent prior to participation. Because the study posed minimal risk and was conducted online, the IRB approved a waiver of written documentation of consent. Participants provided digital consent after reading a disclosure statement describing the study's purpose, procedures, and their rights as research participants. No personally identifiable information was collected, and all responses were stored and analyzed in de-identified form. Initially, 1,092 respondents participated, but the final dataset was adjusted to 1,000 participants to align with a sampling frame based on sex, age, race, and education levels. This sampling frame was developed using data from multiple national sources, including the American Community Survey (ACS), public voter records, the 2020 Current Population Survey (CPS) Voting and Registration supplements, the 2020 National Election Pool (NEP) exit poll, and the 2020 Cooperative Election Study (CES) surveys. The inclusion of demographic characteristics and voting history from the 2020 U.S. presidential election helped ensure representativeness. The survey results have a margin of error of ±3.38 percentage points. Participants were recruited from YouGov's extensive panel of 1.8 million U.S. residents, with efforts to enhance diversity and representativeness through various outreach methods, including web advertising, email invitations, telephone solicitations via random digit dialing, and mail-based recruitment using random address selection (YouGov, 2020). While this study relied on an opt-in panel with demographic weighting to approximate the general U.S. population, there remains a small potential for selection bias despite these measures.

### Sample characteristics

The average age of respondents was 49 years (SD = 17.71), spanning 19–88 years. Participants were asked to select all the biological sex and gender categories that applied to them, so the sex and gender categories do not total 1,000. The sample included 480 males, 510 females, 4 non-binary/genderqueer individuals, 5 transgender respondents, 2 participants who opted not to disclose their biological sex or gender, and 1 person identifying as "no label." Educational background varied, with 346 respondents (approximately one-third) holding a four-year college degree or higher, while 654 participants had not completed a four-year degree. Income distribution across the sample included 267 lower-income participants (earning under $30,000 per year), 443 middle-income participants ($30,000–$99,999 per year), and 290 higher-income participants (earning $100,000 or more annually). The racial composition was 64.7% White, 12.1% Black or African American, 14.8% Hispanic or Latino, 3.3% Asian, and 5.1% Native American, Middle Eastern, or other racial/ethnic backgrounds. Geographic factors were also assessed. Using respondents' zip codes, population density was determined, ranging from 3 to 125,860 people per square mile, with an average of 5,185 people per square mile and a median of

1,425 people per square mile. In terms of urban-rural classification, 85.1% of participants resided in metropolitan areas (central counties with at least 50,000 residents in urban areas), while 14.8% lived in non-metro regions (areas with fewer than 50,000 people). Table 1 shows key demographic comparisons between our study sample and the US population (US Census Bureau, 2025).

## Survey procedures

Upon opening the survey, participants were asked questions about their general science and technology beliefs (those questions were not used in the current analysis), given a brief description of microbiome engineering in the built environment (see Supplemental Materials) and then asked questions about their general perceptions of microbiome engineering (see [11,16]). Next, participants were asked to read the following description of a potential use of microbiome engineering: "Hospitals face a difficult challenge to stop harmful microbes that can lead to infections. Typical treatments are not always effective because some microbes have become resistant, meaning they don't respond to medicines or treatments. As a solution, there is growing interest in the idea to purposefully introduce microbiomes in hospital sinks to support the growth of good microbes and control the growth of harmful ones." Finally, participants answered questions about their attitudes of introduced microbes in hospital sinks.

## Independent variables

In addition to demographic and sociographic variables, the study incorporated independent variables identified in prior research as predictors of support for microbiome engineering in the built environment [11,16]. Political party identification was measured using a 7-point scale (strong Republican coded high and strong Democrat coded low), with a mean score of 3.82 (SD = 2.21). Church attendance was measured by asking participants how often they attended church with response options of 1 ('never'), 2 ('seldom'), 3 ('a few times a year'), 4 ('once or twice a month'), 5 ('once a week'), or 6 ('more than once a week'), with a mean of 2.78 (SD = 1.77).

Four general attitudes about microbiome engineering (ME) were also included as independent variables. Prior information-seeking about ME was measured with one question ($M = 2.25$, $SD = 1.10$), familiarity with ME was assessed through a composite score (the average) of three questions ($M = 2.57$, $SD = .96$, Cronbach's $\alpha = .83$), perceived knowledge of ME was assessed as a composite variable with four questions ($M = 2.71$, $SD = .92$, Cronbach's $\alpha = .84$), and desire to learn more about ME was assessed with one item ($M = 3.59$, $SD = 1.07$). See Supplemental Materials for the full question wording of these items and all the items described below.

Emotions about ME served as another set of predictors. Participants were asked to respond to seven negative emotions and seven positive emotions on a scale from 1 (not at all) to 5 (very high) immediately after reading the initial description of microbiome engineering. A composite variable of negative emotions was created by averaging scores to the seven negative emotions (disgust, distaste, anger, fear, anxiety, helplessness, and nervousness; $M = 1.96$, $SD = .90$, Cronbach's $\alpha = .92$). Likewise, a composite variable of positive emotions was created (excitement, satisfaction, relief, enthusiasm, optimism, wonder, and contentment; $M = 2.49$, $SD = .92$, Cronbach's $\alpha = .91$).

**Table 1. Comparison of Sample Demographics and US Census Demographics.**

|  | Sample | Census Estimate |
| --- | --- | --- |
| Age (65 years and older) | 23.3% | 17.7% |
| Sex (Female) | 51% | 50.5% |
| Race (White, not Hispanic) | 64.7% | 58.4% |
| Education (4-year college degree or higher) | 34.6% | 35.0% |
| Median household income range | $50,000-$59,000 | $70,000-$79,000 |

Both threat and coping appraisals were included as independent variables. Threat appraisal of antimicrobial resistance and hospital-acquired infections was measured with two variables: a severity perception ($M = 3.68$, $SD = .91$) and a susceptibility perception ($M = 3.39$, $SD = .96$). Coping appraisal of introduced microbes (IM) was measured with three items: response efficacy perception of IM in hospital sinks ($M = 3.10$, $SD = 1.03$), trust in hospital management of IM ($M = 3.10$, $SD = 1.09$), and trust in healthcare regulatory agencies to provide clear guidelines for the safe use of IM ($M = 3.25$, $SD = 1.13$).

Two items that represented secondary risk perception of IM in hospital sinks served as a final set of predictors. The two items addressed concerns about potential risks and safety issues with IM ($M = 3.64$, $SD = .97$) and unintended consequences of IM on water quality and health ($M = 3.70$, $SD = .95$).

### Dependent variables

Three dependent variables were used to answer the study's research question. The first dependent variable (Model 1), *support for IM in hospital sinks to reduce AMR and HAIs,* was an average of two items, "I am open to the idea of using introduced microbiomes if they are shown to be effective in reducing microbial resistance," and "I would support the implementation of introduced microbiome solutions in hospital sinks if they were part of a comprehensive strategy to combat microbial resistance," ($M = 3.43$, $SD = 1.06$, $r = .714$). The second dependent variable (Model 2), *support for rigorous evaluation and testing of IM,* was measured with a single item, "I think that the introduction of microbiomes in hospital sinks should be subject to rigorous testing and evaluation," ($M = 4.14$, $SD = .91$). The third dependent variable (Model 3), *support healthcare staff and patient education of IM,* was also measured with a single item, "I believe that healthcare staff and patients should be educated about the use and benefits of introduced microbiomes in sinks," ($M = 4.15$, $SD = .88$).

### Analytic approach

To examine the factors influencing public support for the introduction of beneficial microbes in hospital environments, we employed ordinary least squares (OLS) hierarchical regression analyses in SPSS version 29. This analytical approach allowed us to assess the incremental contribution of different sets of predictors to attitudes toward microbiome interventions while controlling for potential confounding variables. The regression models were constructed using a block-wise entry method, with variables entered in conceptually informed stages to reflect their presumed causal order. The first block included basic demographic and sociographic variables: age, sex (measured as 1 = male, 0 = non-male), race, education, income, and metro status), along with political party identification and church attendance. These measures captured individual differences in values and worldview orientations that often shape attitudes toward science and health policy. The second block focused on general engagement with ME including prior information-seeking, familiarity, perceived knowledge, and desire to learn more. These constructs reflect respondents' baseline exposure and motivation to understand ME, which are expected to influence openness to its application. In the third block, we introduced affective responses, incorporating both positive and negative emotions toward ME, based on evidence that emotional reactions significantly influence risk perception and public evaluations of emerging technologies. The fourth block captured threat appraisal related to AMR and HAIs, assessing perceived severity and susceptibility—key motivational elements from health behavior theories (Cummings, Rosenthal, and Kong, 2020). The fifth block included coping appraisal variables, specifically beliefs in the efficacy of IM in hospital sinks, as well as trust in hospital management and healthcare regulatory agencies to safely oversee implementation. These items address institutional trust and perceptions of effectiveness, which are known to affect public acceptance of risk mitigation interventions. Finally, the sixth block added variables capturing secondary risk perceptions, such as concerns about safety, unintended health impacts, and environmental consequences of introducing microbes into clinical settings. This structure enabled us to observe how each successive block contributed to explaining support for IM, allowing for a nuanced interpretation of the psychological, social, and informational drivers of public opinion.

## Results

The regression models revealed several meaningful predictors of public support for IM interventions in hospital environments. Model 1 examined predictors of support for introducing microbiomes into hospital sinks to combat antimicrobial resistance and hospital-acquired infections. Across all blocks, the final model explained 57.7% of the variance in support for introducing IM into hospital sinks, suggesting a substantial portion of public opinion on this issue is shaped by both individual dispositions and cognitive-emotional responses to microbiome technologies. Demographic, sociographic, and value-related variables in the first block explained 10.6% of the variance. Among these, identifying as white was positively associated with support for IM ($\beta = .068$, $p = .003$), as was having a four-year college degree ($\beta = .060$, $p = .011$). Political party identification was also significant, with more Republican respondents expressing less support ($\beta = -.071$, $p = .002$). Age, biological sex, metro area status, annual income, and church attendance were not significant predictors. In the second block, general engagement with microbiome engineering (ME) added another 17.5% of explained variance. The strongest predictor in this block was the desire to learn more about ME ($\beta = .171$, $p < .001$), while familiarity with microbiome engineering (ME) was negatively associated with support ($\beta = -.120$, $p = .006$), possibly reflecting that increased familiarity is associated with more critical evaluations. Information-seeking behavior and perceived knowledge were not statistically significant. Emotional responses to ME, introduced in the third block, accounted for an additional 11.1% of the variance. Negative emotion toward ME significantly predicted lower support ($\beta = -.167$, $p < .001$), while positive emotion predicted higher support ($\beta = .092$, $p = .002$), confirming the strong role of affect in shaping public attitudes toward science and technology support. Threat appraisal variables in block four contributed minimally, with only perceived severity of AMR and HAIs reaching significance ($\beta = .051$, $p = .030$), while perceived susceptibility was not significant. The fifth block, coping appraisal, contributed the largest single increase in explanatory power (18.3%). Belief in the response efficacy of IM in hospital sinks was the strongest predictor in the full model ($\beta = .320$, $p < .001$). Trust in hospital management ($\beta = .093$, $p = .003$) and in healthcare regulatory agencies ($\beta = .227$, $p < .001$) were also positively associated with support. Finally, block six, which included concerns about risks and unintended consequences of IM, did not contribute to any significant additional variance, and none of the risk perception variables reached statistical significance. These results suggest that support for IM in hospital sinks is most strongly influenced by more positive (and less negative) emotional responses to ME, trust in institutions, perceived response efficacy, and a desire to engage with microbiome science, rather than by perceived risks or general demographic factors alone.

Model 2 explored the predictors of support for rigorous evaluation and testing IM prior to their implementation in hospital settings. This model captured a different dimension of public attitude, one focused less on general favorability of the scientific intervention and more on procedural caution and oversight. The full model explained 24.8% of the variance in support for evaluation and testing, with several statistically significant predictors emerging across blocks. In the first block, demographic and sociographic factors accounted for 5.4% of the variance. Identifying as white was again positively associated with support ($\beta = .118$, $p < .001$), and more Republican identification was negatively associated ($\beta = -.076$, $p = .015$). Age was also a significant predictor in this model ($\beta = .091$, $p = .004$), indicating that older respondents were more likely to support thorough evaluation procedures. Other variables, including sex, metro area status, education, income, and church attendance, were not significant. The second block, which included general attitudes and behaviors related to microbiome engineering (ME), contributed an additional 7.4% of explained variance. Prior information-seeking about ME was negatively associated with support for testing ($\beta = -.200$, $p < .001$), as was familiarity with ME ($\beta = -.159$, $p = .006$), potentially reflecting a tension between perceived knowledge and regulatory caution. Conversely, both perceived knowledge ($\beta = .128$, $p = .029$) and desire to learn more about ME ($\beta = .104$, $p = .002$) were positively associated with support for evaluation, suggesting that curiosity and a willingness to engage with scientific complexity are linked to preferences for procedural rigor. Emotional responses added little explanatory power (0.3%), though both negative emotion ($\beta = -.070$, $p = .033$) and positive emotion ($\beta = .105$, $p = .007$) reached significance, suggesting that individuals who have less negative emotion and more positive emotion demand accountability. Block four, which addressed threat appraisals of AMR and HAIs, added

2.3% of variance. Perceived severity was a significant predictor (β = .090, p = .004), reinforcing the idea that heightened concern about the magnitude of AMR encourages support for careful evaluation of microbial interventions. Perceived susceptibility did not significantly predict support. Coping appraisal variables in block five had a limited effect, adding only 0.5% of variance. Belief in the efficacy of IM was positively associated with support for testing (β = .133, p = .002), though trust in hospital management and healthcare regulatory agencies were not significant in this context. These findings suggest that belief in the intervention's effectiveness may coexist with an expectation for strong regulatory oversight. Finally, secondary risk perception variables in block six contributed a substantial 8.9% of the model's explanatory power. Concerns about the potential safety risks of IM (β = .147, p < .001) and worries about unintended environmental or health consequences (β = .247, p < .001) were both significant predictors. This highlights that individuals who express support for testing are particularly motivated by concerns about unknown or poorly understood risks, even if they also believe in the intervention's potential. Taken together, Model 2 demonstrates that support for the rigorous evaluation of IM is shaped by a combination of demographic predispositions, informational engagement, more positive emotional responses to ME, and especially risk perception. Unlike support for implementation (Model 1), which was more strongly driven by trust and efficacy beliefs, this outcome is grounded more firmly in caution, concern for unintended consequences, and a desire for scientific transparency.

Model 3 examined predictors of support for educational initiatives aimed at healthcare staff and patients regarding the use of introduced microbes (IM) in hospital settings. This model captured how individuals evaluate the importance of communication, training, and knowledge dissemination as a strategy for safe and acceptable implementation of microbiome interventions. The full model accounted for 23.9% of the variance in support for IM-related education, reflecting a moderate but meaningful relationship between individual characteristics and support for these initiatives. The first block, encompassing demographic, sociographic, and value-oriented variables, explained 3.8% of the variance. Age emerged as a positive predictor (β = .106, p < .001), while sex was negatively associated with support (β = −.100, p < .001), indicating that older respondents and females were more likely to endorse education and training around IM technologies. Other variables in this block, including race, metro area status, education, income, political party identification, and church attendance—were not significant predictors of educational support. The second block, focused on general attitudes and behaviors related to microbiome engineering (ME), added 11.1% of explanatory power. Respondents who had engaged in prior information-seeking about ME were less supportive of education efforts (β = −.146, p < .001), and a similar negative association was observed for familiarity with ME (β = −.125, p = .033). These findings mirror the earlier models and suggest a potential skepticism among those who feel more informed or already engaged with the topic. In contrast, a strong desire to learn more about ME was a significant positive predictor (β = .234, p < .001), reinforcing the idea that educational enthusiasm is tied to support for institutional knowledge-sharing efforts. Perceived knowledge was not a significant predictor. Emotional responses added just 0.4% of additional variance. Negative emotion toward ME was significantly associated with lower support for education (β = −.090, p = .007), while positive emotion was not significant. These findings suggest that negative affect toward ME may deter support for broader communication efforts, even in the absence of strong opposition to IM itself. Block four, which addressed perceived threat from AMR and HAIs, contributed only 0.6% of the variance. Neither perceived severity nor susceptibility was significantly associated with support for education, suggesting that threat-based motivations may play a less prominent role in public evaluations of informational initiatives compared to support for the IM interventions. Coping appraisal variables in block five added 1.0% of variance. Belief in the efficacy of IM in hospital sinks significantly predicted greater support for staff and patient education (β = .161, p < .001), whereas trust in hospital management and healthcare regulatory agencies did not reach significance in this model. This implies that support for educational strategies is more strongly tied to confidence in the intervention itself than to institutional governance. The final block, which included secondary risk perceptions, contributed 7.0% to the model's explanatory power. Concerns about the potential risks of IM (β = .178, p < .001) and worries about unintended consequences (β = .173, p < .001) were both strongly associated with support for education. These suggest that individuals who perceive higher risks are more

likely to advocate for informational interventions, potentially as a way to mitigate uncertainties or enhance transparency. Altogether, Model 3 demonstrates support for educational initiatives surrounding IM is driven by age, biological sex, curiosity, perceived efficacy, and especially concern about unintended risks. While emotional and threat-based responses played a limited role, the desire to learn more and the perception of risk emerged as critical motivators for public endorsement of education aimed at healthcare professionals and patients. Table 2 below reports the full findings of all 3 models and Fig 1 lists the significant predictors for each model.

## Discussion

This study reports the first nationally representative analyses of U.S. public perceptions of microbiome engineering in hospital settings, focusing on three key dimensions: support for implementation (Model 1), support for rigorous evaluation and testing (Model 2), and support for staff and patient education (Model 3). Collectively, these models reveal distinct but overlapping predictors of public engagement with ME in the context of HAI prevention. The findings highlight that while cognitive and demographic factors are relevant, attitudes are most strongly shaped by emotional responses, trust in institutional efficacy, perceived risks and benefits, and a desire to engage with microbiome science. Across all three models, support was consistently associated with a desire to learn more about microbiome engineering, underscoring the role of intellectual curiosity as a proxy for openness and engagement. Emotional reactions, particularly negative emotions, played a suppressive role, while perceived efficacy and trust in oversight mechanisms were crucial for fostering support for both implementation and oversight. Interestingly, prior information-seeking and familiarity with ME were often negatively associated with support. One possible explanation is that surface-level or critically framed exposure, such as through media emphasizing risks, may contribute to public skepticism, perhaps due to the conflation of microbiome engineering with genetic modification or synthetic biology. While this remains a hypothesis, it highlights the need for further research into how the framing and context of information shape public attitudes toward emerging biotechnologies. Risk perceptions also played a more significant role in shaping support for evaluation and education than in support for direct implementation, indicating that members of the public may differentiate between endorsing a technology in principle and advocating for safeguards in its real-world deployment. Interpreting these findings through the lens of the RRI framework provides insight into how biotechnology developers, policymakers, and hospital stakeholders can align microbiome interventions with societal values. The four core RRI principles, anticipation, inclusion, reflexivity, and responsiveness, offer a normative guide for operationalizing the implications of each model.

Findings from Model 1 indicate that support for the direct use of introduced microbiomes in hospital environments is shaped most strongly by beliefs in the efficacy of the intervention and trust in regulatory institutions. Additionally, emotional responses, both positive and negative, significantly influenced support, suggesting that affective framing will be key in shaping public opinion. From an RRI perspective, these findings stress the importance of anticipation and responsiveness. Biotechnology developers and hospital leaders must anticipate emotional reactions and public concerns, not just technical risks, and embed these insights early in development and implementation. Demonstrating intervention efficacy transparently, coupled with proactive communication strategies that address emotional dimensions, will be crucial for building public trust. Policy makers should ensure that regulatory oversight mechanisms are not only robust but also perceived as transparent and adaptive, fostering public confidence in their ability to manage both benefits and unintended consequences. Hospital administrators should consider pilot programs that showcase efficacy and safety, potentially with third-party evaluations, to build credibility and demonstrate responsiveness to public concern.

In Model 2, support for oversight and testing was most strongly predicted by perceived risk—particularly concerns over unintended consequences—as well as information engagement and severity perception of AMR. Here, the RRI principles of reflexivity and inclusion are most salient. The public's desire for caution reflects a recognition of the complex, uncertain, and context-dependent nature of emerging microbiome engineering. Developers must therefore engage in reflexive practices, interrogating their assumptions, risk models, and communication approaches. This includes making space for public

**Table 2. Hierarchical regression models for (1) support for introduced microbiomes (IM) in hospital sinks and (2) support for oversight and education for IM in hospital sinks.**

| Item | Model 1: Support for IM in hospital sinks | | Model 2: Support for rigorous evaluation and testing of IM | | Model 3: Support for healthcare staff and patient education of IM | |
|---|---|---|---|---|---|---|
| | Stand. β | p-value | Stand. β | p-value | Stand. β | p-value |
| Demographics, sociographics, and value dispositions | | | | | | |
| Age | .001 | .952 | .091 | .004** | .106 | <.001*** |
| Biological sex (male coded high) | .013 | .545 | −.002 | .950 | −.100 | <.001*** |
| Race (white = 1; nonwhite = 0) | .068 | .003** | .118 | <.001*** | .038 | .201 |
| Metro Area Status (metro = 1; nonmetro = 0) | −.010 | .637 | .029 | .317 | −.001 | .964 |
| Highest level of education (1 = four-year college degree; 0 = no college degree) | .060 | .011** | .037 | .235 | .004 | .909 |
| Annual income (higher income coded high) | .031 | .179 | .048 | .119 | .049 | .116 |
| Political party identification (Republican coded high) | −.071 | .002** | −.076 | .015** | .001 | .985 |
| Church attendance (more frequent coded high) | .005 | .825 | −.003 | .925 | .002 | .960 |
| Block 1 R² (%) | 10.6%*** | | 5.4%*** | | 3.8%*** | |
| General attitudes and behaviors about ME | | | | | | |
| Prior information-seeking about ME | −.043 | .170 | −.200 | <.001*** | −.146 | <.001*** |
| Familiarity with ME | −.120 | .006** | −.159 | .006** | −.125 | .033* |
| Perceived knowledge and understanding about ME | .047 | .281 | .128 | .029* | .071 | .231 |
| Desire to learn more about ME | .171 | <.001*** | .104 | .002** | .234 | <.001*** |
| Block 2 R² change (%) | 17.5%*** | | 7.4%*** | | 11.1%*** | |
| Emotions toward ME | | | | | | |
| Negative emotion toward ME | −.167 | <.001*** | −.070 | .033* | −.090 | .007** |
| Positive emotion toward ME | .092 | .002** | .105 | .007** | .035 | .369 |
| Block 3 R² change (%) | 11.1%*** | | 0.3% | | 0.4% | |
| Threat appraisal of AMR HAIs | | | | | | |
| Severity (magnitude) perception | .051 | .030* | .090 | .004** | .008 | .798 |
| Susceptibility (likelihood) perception | .022 | .338 | −.008 | −.274 | .012 | .685 |
| Block 4 R² change (%) | 0.2% | | 2.3%*** | | 0.6%* | |
| Coping appraisal of IM | | | | | | |
| Response efficacy of IM in hospital sinks | .320 | <.001*** | .133 | .002** | .161 | <.001*** |
| Trust in hospital management of IM | .093 | .003** | .005 | .901 | .026 | .535 |
| Trust in healthcare regulatory agencies to provide clear guidelines for the safe use of IM | .227 | <.001*** | .070 | .081 | .072 | .076 |
| Block 5 R² change (%) | 18.3%*** | | 0.5% | | 1.0%** | |
| Secondary risk perception of IM | | | | | | |
| Concerns about potential risks and safety issues with IM | −.008 | .781 | .147 | <.001*** | .178 | <.001*** |
| Concerns with unintended consequences of IM on water quality or health | .019 | .510 | .247 | <.001*** | .173 | <.001*** |
| Block 6 R² change (%) | 0.0% | | 8.9%*** | | 7.0%*** | |
| Total R² (%) | 57.7%*** | | 24.8% | | 23.9%*** | |

Note. The R² change is provided for each set of predictors, and standardized coefficients for the final model are listed. R² change was deemed significant when the F-change test was significant (p < .05). * p < .05, ** p < .01, *** p < .001.

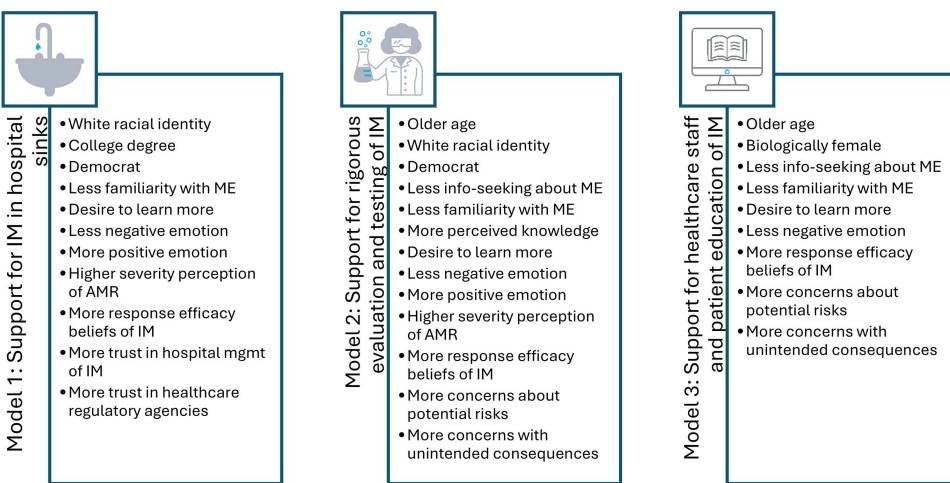

**Fig 1. Significant predictors are listed respectively for each model.**

skepticism—not dismissing it—as a valid form of engagement. Policy makers should formalize participatory evaluation processes that allow lay publics and healthcare workers to shape evaluation criteria and flag unintended impacts. Furthermore, inclusion mandates that hospital stakeholders incorporate not only expert input but also frontline staff and patients into oversight design, ensuring the measures reflect diverse concerns and do not default to technocratic rationality. By doing so, stakeholders can move from a compliance-based model of safety assurance to a trust-building model of co-evaluation.

Model 3 shows that support for staff and patient education was shaped by perceived risk, desire to learn, age (older people are more supportive), sex (females are more supportive), and belief in the intervention's efficacy. The consistent positive association with curiosity (desire to learn more) suggests strong public interest in participatory understanding of microbiome technologies, while negative emotion and prior familiarity predicted lower support—perhaps indicating resistance rooted in concern rather than apathy. This model foregrounds the RRI principles of inclusion and reflexivity once again but also emphasizes responsiveness. Biotechnology developers must design educational tools that go beyond technical knowledge to include emotional engagement, cultural framing, and historical contexts that have shaped public skepticism toward engineered microbes. This could include storytelling, analogies, and co-designed materials with healthcare workers and patients. Policy makers and funders should support training programs for clinicians that not only teach microbial literacy but also cultivate their capacity as knowledge brokers who can translate and contextualize microbiome interventions to patients. For hospital systems, embedding education as a co-learning process—where staff and patients jointly explore the implications of microbiome innovations—can enhance buy-in, identify unanticipated challenges, and foster a culture of shared accountability. Additionally, education may serve as a means of risk communication, building public resilience and trust in microbiome-based interventions through transparency and empowerment. Together, these findings reinforce that microbiome interventions are not merely technical challenges, but deep social and ethical endeavors. The RRI framework provides an invaluable roadmap for guiding not only public engagement but also institutional design and innovation strategy. Ensuring that implementation, evaluation, and education efforts are anticipatory, inclusive, reflexive, and responsive will be key to securing a socially robust path forward for microbiome engineering in hospitals. These principles can help avoid the pitfalls seen in earlier biotechnology controversies and foster durable public trust in a new generation of microbiome engineering interventions.

An important advantage in the current stage of microbiome-based interventions is that the technology remains in early development, creating a valuable window for thoughtful, anticipatory governance. In the short term, several concrete steps

can help align ME innovation with RRI principles and foster public trust. First, biotechnology developers should prioritize interdisciplinary collaborations that include social scientists, ethicists, infection control experts, and public engagement specialists to co-design communication and evaluation strategies. Second, pilot education programs can be developed and tested within a small number of hospital settings, particularly those already exploring microbiome-informed infection control, to better understand how patients and staff respond to various framing, delivery methods, and cultural contexts. Third, policy makers and funding agencies should invest in the creation of transparent testing and feedback mechanisms, including publicly accessible trials and regulatory dashboards that communicate emerging risks, successes, and lessons learned. Finally, hospital administrators can create advisory committees that include patients, frontline healthcare workers, and infection prevention staff to weigh in on readiness, training needs, and oversight measures. These short-term steps not only allow developers to refine ME technologies and their implementation processes, but also demonstrate a proactive, values-based commitment to responsible innovation—one that positions microbiome engineering as not just scientifically sound, but socially legitimate and ethically grounded.

This study has several limitations that should be noted. First, while the sample was nationally representative on several key demographic indicators, it included a slightly higher proportion of respondents over age 65 and individuals identifying as white compared to recent U.S. Census estimates. These demographic differences may influence generalizability, particularly in relation to age- and race-related variation in trust, health experience, or familiarity with emerging biotechnologies. Second, the survey did not include healthcare workers, whose frontline experiences and institutional perspectives are likely to shape attitudes toward microbiome interventions in distinct ways. Future work should examine how perceptions vary between the general public and clinical stakeholders. Third, we did not collect data on participants' personal hospital histories (e.g., recent or frequent admissions, experiences with HAIs, or the hospitalization of loved ones), which may significantly shape their emotional responses or perceived vulnerability. Incorporating these experiential variables in future research could enhance understanding of how direct exposure to healthcare environments influences support for microbiome-based innovations. Future study could also consider targeted sampling of often-underrepresented populations, including Black, African American, and Latinx communities, to explore how culturally specific experiences and historical health disparities shape perceptions of microbiome engineering and hospital-based interventions.

## Conclusion

This study reports a comprehensive examination of public attitudes toward microbiome engineering in hospital environments, contributing new empirical insight into the factors shaping support for their implementation, oversight, and education. Drawing on a nationally representative sample of U.S. adults and guided by the RRI framework, the findings reveal that public support for introduced microbiomes is most strongly influenced by emotional responses, trust in institutional oversight, perceived intervention efficacy, and a desire to learn. While general demographic factors played a modest role, cognitive engagement with microbiome science, particularly the motivation to understand it better—emerged as a consistent and powerful predictor of support. Moreover, support for evaluation and education efforts was notably driven by concerns about safety and unintended consequences, underscoring the importance of transparency and inclusive governance. Together, these results suggest that efforts to responsibly develop and deploy IM in healthcare settings must account for not only scientific and technical efficacy but also public values, emotional responses, and social trust.

Looking ahead, the relatively early stage of IM development presents a critical opportunity for stakeholders to proactively shape its trajectory in alignment with societal expectations and ethical considerations. Future success will depend on embedding RRI principles into the full life cycle of microbiome engineering—from research design and testing to clinical deployment and public education. Developers must adopt reflexive, inclusive approaches that welcome diverse perspectives and anticipate long-term consequences. Policymakers and hospital systems can play pivotal roles by supporting transparent evaluation frameworks and participatory education strategies that empower staff and patients alike. As biotechnology continues to advance, microbiome-based innovations in hospitals offer a chance not only to address urgent

challenges like antimicrobial resistance, but also to reimagine how science, health, and society interact. By embracing responsibility from the outset, stakeholders can ensure that microbiome engineering technologies are not only effective, but also equitable, trusted, and ethically grounded in the values of the communities they are intended to serve.

## Disclaimer

The funders had no role in study design, data collection and analysis, decision to publish, or preparation of the manuscript. The views and opinions expressed in this work are those of the authors and do not necessarily reflect the official policy or position of any affiliated institutions, organizations, or agencies.

## Supporting information

**S1 File. Supplementary Materials.** https://example.com/file1-supplementary-materials.
(DOCX)

**S2 File. IRB exemption.**
(PDF)

## Author contributions

**Conceptualization:** Christopher L. Cummings, Jennifer Kuzma.

**Data curation:** Christopher L. Cummings, Jennifer Kuzma.

**Formal analysis:** Christopher L. Cummings, Kristen D. Landreville, Jennifer Kuzma.

**Funding acquisition:** Jennifer Kuzma.

**Investigation:** Christopher L. Cummings, Kristen D. Landreville, Jennifer Kuzma.

**Methodology:** Christopher L. Cummings, Kristen D. Landreville, Jennifer Kuzma.

**Project administration:** Christopher L. Cummings, Kristen D. Landreville, Jennifer Kuzma.

**Resources:** Jennifer Kuzma.

**Software:** Christopher L. Cummings, Kristen D. Landreville, Jennifer Kuzma.

**Supervision:** Kristen D. Landreville, Jennifer Kuzma.

**Validation:** Christopher L. Cummings, Kristen D. Landreville, Jennifer Kuzma.

**Visualization:** Christopher L. Cummings, Kristen D. Landreville, Jennifer Kuzma.

**Writing – original draft:** Christopher L. Cummings, Kristen D. Landreville, Jennifer Kuzma.

**Writing – review & editing:** Christopher L. Cummings, Kristen D. Landreville, Jennifer Kuzma.

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
