## [Decision Letter · Decision Letter 0]

8 Jul 2025

PONE-D-25-30050Public Perceptions and Support for Introduced Microbes to Combat Hospital-Acquired Infections and Antimicrobial ResistancePLOS ONE

Dear Dr. Cummings,

Thank you for submitting your manuscript to PLOS ONE. After careful consideration, we feel that it has merit but does not fully meet PLOS ONE’s publication criteria as it currently stands. Therefore, we invite you to submit a revised version of the manuscript that addresses the points raised during the review process. Please submit your revised manuscript by Aug 22 2025 11:59PM. If you will need more time than this to complete your revisions, please reply to this message or contact the journal office at plosone@plos.org . Please include the following items when submitting your revised manuscript:

We look forward to receiving your revised manuscript.

Kind regards,

Assoc. Prof. Phakkharawat Sittiprapaporn, Ph.D.

Academic Editor

PLOS ONE

Journal Requirements:

“National Science Foundation”

6. We note that you have indicated that there are restrictions to data sharing for this study. For studies involving human research participant data or other sensitive data, we encourage authors to share de-identified or anonymized data. However, when data cannot be publicly shared for ethical reasons, we allow authors to make their data sets available upon request. For information on unacceptable data access restrictions, please see http://journals.plos.org/plosone/s/data-availability#loc-unacceptable-data-access-restrictions.

8. We note that Figure 1 in your submission contain copyrighted images. All PLOS content is published under the Creative Commons Attribution License (CC BY 4.0), which means that the manuscript, images, and Supporting Information files will be freely available online, and any third party is permitted to access, download, copy, distribute, and use these materials in any way, even commercially, with proper attribution. For more information, see our copyright guidelines: http://journals.plos.org/plosone/s/licenses-and-copyright.

Reviewers' comments:

Reviewer's Responses to Questions

**Comments to the Author**

1. Is the manuscript technically sound, and do the data support the conclusions?

Reviewer #1: No

Reviewer #2: Yes

Reviewer #3: Yes

2. Has the statistical analysis been performed appropriately and rigorously? 

Reviewer #1: No

Reviewer #2: Yes

Reviewer #3: Yes

3. Have the authors made all data underlying the findings in their manuscript fully available?

Reviewer #1: Yes

Reviewer #2: No

Reviewer #3: Yes

4. Is the manuscript presented in an intelligible fashion and written in standard English?

Reviewer #1: No

Reviewer #2: Yes

Reviewer #3: Yes

5. Review Comments to the Author

Reviewer #1: The work by Cummings and collogues encapsulates an interesting investigation of the factors influencing one’s attitudes to using microbes as a preventative in preventing hospital acquired infections. There are many errors in the introduction where papers are cited that do not accurately support their claims. This reduces the reviewer’s enthusiasm for the work and their trust in the author’s presented results.

- Personally, I would reduce the number of abbrevations. It makes the text overly verbose. Too many to keep track of.

- Reference to Rodrigues et al 2025 does not support using engineered bacteria, at no point in the cited study were probiotics used, however it was noted that women who were hospitalized were at a higher risk of acquiring an infection.

- Reference to Fady once again does not support the claim being made "For example, bioengineered waste lignocellulosic biochars have been shown to sequester MDR bacteria, reducing hospital contamination and minimizing antibiotic- resistant outbreaks (Fady et al., 2024).”

- This paper is not about phages at all. Discusses nanoparticles to treat infections but does not mention either of the points it is meant to support “Additionally, phage therapy and microbiome restoration techniques have gained traction as alternative interventions, showing promise in targeting antimicrobial-resistant bacterial strains without relying on traditional antibiotics (Ahmed et al., 2025).” Phage therapy has existed as a concept since the early 1900s, there should be at least a few references which would be better used to support this claim.

- Another instance of an improper citation: “Recurrent bacteremia in the setting of Pseudomonas endocarditis of the tricuspid valve and indwelling medical devices”, cited work is case report, regarding biofilm formation on a medical device. No review of supplying probiotics to outcompete pathogens as suggested by the quoted line.

- There is no Cummings 2025 in the works cited. I assume that this is in reference to the manuscript itself, which seems odd if not misleading: “So far there have been limited studies of U.S. public attitudes on microbiome engineering in the built environment. In the first survey of built microbiome engineering and public attitudes, attitudes toward microbes and microbiome engineering were discovered to be shaped by a complex interplay of scientific literacy, emotional responses, media influence, and demographic factors (Cummings, Landreville, & Kuzma, 2024; Cummings, Landreville, & Kuzma, 2025)”.

- It would be interesting to see if one’s prior scientific knowledge was in anyway predictive of their thoughts on the microbiome in therapy, perhaps including this information in the model would provide interesting insights as to whether education is actually helpful in increasing the public’s perception of microbiome based therapeutics

Reviewer #2: Cummings and colleagues present a very thorough, well written, and interesting assessment of Americans’ attitudes towards introducing engineered microbiomes into hospital sinks to combat antimicrobial resistance and hospital associated infections. The study is timely as AMR and HAI are major issues facing the US healthcare system and innovative strategies are needed to combat them. Additionally, deploying engineered microbiomes in the built environment is in early stages, making now an ideal time for assessing public attitudes so that they may inform future implementation strategies and promote success. I think the manuscript can be improved by adding some detail/clarifying information, tightening up some of the language to describe survey respondent characteristics, and including a discussion of the study’s limitations. My specific suggestions are below.

Major comments

1. I think it would be nice to give a little detail/context on why the study/survey focused on introduced microbiomes specifically deployed in sinks (as opposed to other parts of the hospital built environment).

2. I think the last sentence of the introduction – very clearly stating the overall research question – is really great and sets the reader up nicely for the rest of the manuscript! (no revisions requested here)

3. Are there any notable examples/success stories of deploying engineered/introduced microbiomes in other built environment settings (healthcare or otherwise)? If yes, I would consider giving a brief overview of these in the literature review section, along with any lessons learned that may be relevant to the current study. Alternatively, are there any notable failures with relevant lessons learned?

4. The language used to describe sex/gender, political orientation/party identification, and religiosity/church attendance is not internally consistent throughout the manuscript, and the alternative terms that are used have meaningful differences. Here are some specific clarifying questions and suggestions I have:

a. I really like that you give a full description of all participants’ reported gender in the methods! From those data, it seems like the survey asked about gender and not sex? If yes, terminology used later in the manuscript to refer to ‘females’ and ‘biological sex’ should be revised to refer to women and gender. If the survey assessed gender and sex, that should be stated in the methods, and the manuscript should clearly identify which variable is being discussed and use congruent terminology (e.g., ‘female’ to describe sex and ‘woman’ to describe gender).

b. The methods section defines the bounds of the party identification scale as ‘strong Republican coded’ and ‘strong Democrat coded’. Later, the results refer to ‘conservative political orientation’. In the current (and recent) American sociopolitical context, ‘Republican’ and ‘conservative’ are not interchangeable, and I think whichever language is intended to be used with the party identification scale should be used throughout the manuscript.

c. Did the church attendance scale explicitly assess ‘church’ attendance, or did it ask about attending any variety of religious/worship service or any variety of house of worship? If it only asked about church, I don’t think ‘religiosity’ or ‘religious attendance’ should be used to describe this independent variable, and it should instead be referred to as church attendance.

5. In the methods or results, can you state how age and gender were operationalized in the models? Specifically, was age continuous or binned? How were individuals who did not share their gender and individuals identifying as nonbinary, genderqueer, trans, and ‘no label’ included in models?

6. Did you examine relationships among the dependent variables? I’m curious if any of the dependent variables are significant predictors of each other.

7. Given the partial overlap of significant predictors between the three models, have you considered including a venn diagram of which ones are/aren’t shared across models, perhaps as a panel b for Figure 1? I think this could be a nice visualization, but also it may get clunky with more text than a typical venn diagram.

8. Can you include a limitations paragraph in the discussion? A couple limitations that I think deserve some discussion are: the survey doesn’t speak to hospital workers’ attitudes (out of the scope of this study but nonetheless important); the survey population has a higher proportion of respondents over 65 and of white respondents than the census estimate; and people’s personal histories related to hospital stays (e.g., recent/frequent/long hospital stays, death of a loved one in hospital) may meaningfully impact their attitudes, but these data were not collected (alternatively, if these data were collected, I think it would be really interesting to include them in the current analysis!).

Minor comments

1. Can you add line numbers in the revised draft?

2. Can you include interquartile ranges in the text alongside standard deviations?

3. There are a few spots where I’d like a bit of clarification:

a. When the methods refer to a ‘composite item’ of multiple questions, does this just mean that response values were averaged across those questions, and the average was used as a predictor?

b. You define the bounds of the 6-point scale used to assess church attendance (1 – rarely attends; 6 – attends weekly). For other scales, it may be easy to assume what the rest of the scale values are defined as, but I’m not sure it’s as intuitive for church attendance. Can you define all six scale values in the methods?

c. Can you state the p value cutoff for determining significance in the methods or results?

d. In the results, does ‘strongest predictor’ refer to the predictor with the largest-magnitude beta estimate or the smallest p value?

e. In the methods or Table 1 footnotes, can you state what test was used to determine significance of changes in R2 across successive blocks of predictors?

f. In the methods, can you state what software programs, packages, functions, and versions were used in the analyses?

4. I’m not sure ‘change’ is the best word to use in Table 1 to describe increases in R2 across successive blocks of predictors. My understanding is that R2 should increase (or stay the same) with each successive block of predictors (i.e., monotonic increases in R2). ‘Change’ implies that R2 may decline with added blocks of predictors, and only presenting the change in R2 for blocks 2-5 may be a bit confusing. I’d suggest either using a different word than ‘change’, or presenting the overall R2 for each block alongside the change from the prior block.

5. The bottom of Figure 1 appears to be cut off.

6. In the first paragraph of the discussion, can you either add a citation for the last part following sentence (starting at ‘potentially’), or state that it is a hypothesis: ‘Interestingly, prior information-seeking and familiarity with ME were often negatively associated with support, suggesting that surface-level or critical exposure may lead to skepticism, potentially due to the framing of microbiome interventions or conflation with genetic modification and synthetic biology.’

Signed: Kayla A. Carter

Reviewer #3: Well-written manuscript highlighting the importance of microbiome engineering to combat HAI. Authors investigated (with an RRI) approach, how people perceive the use of microbes in these "sterile" hospital environments to promote the growth of beneficial microbes.

1. If the participants wanted to know the exact science behind the introduction of beneficial microbes to hospital sinks, were they provided an explanation?

2. Were the surveys completed in person (at a hospital, clinic, at home, etc)?

3. Is it possible to gain more insight into societal value and ethical considerations of microbiome engineering/HAI by surveying individuals in predominately Black, AA and/or Latinx communities?

4. Historically, how influential has the public perception been on developing novel techniques in hospitals?

6. PLOS authors have the option to publish the peer review history of their article (what does this mean? ). If published, this will include your full peer review and any attached files.

**Do you want your identity to be public for this peer review?** For information about this choice, including consent withdrawal, please see our Privacy Policy .

Reviewer #1: No

Reviewer #2: **Yes: ** Kayla A. Carter

Reviewer #3: No

---

## [Author Response · Author response to Decision Letter 1]

5 Aug 2025

Response to Reviewers and Editorial Comments

We thank the editors and reviewers for their insightful comments—we have addressed them all in the revised manuscript and provide granular responses to all comments below. Your feedback has greatly improved the quality and clarity of the work, for which, we are thankful. Best regards,

CC

Editor/Reviewer Comment Author Response

EDITOR

Please ensure that your manuscript meets PLOS ONE's style requirements, including those for file naming. Thank you, will do

Please ensure that you include a title page within your main document. Could you therefore please include the title page into the beginning of your manuscript file itself, listing all authors and affiliations. We have added this.

Please provide additional details regarding participant consent. In the ethics statement in the Methods and online submission information, please ensure that you have specified (1) whether consent was informed and (2) what type you obtained (for instance, written or verbal, and if verbal, how it was documented and witnessed). If your study included minors, state whether you obtained consent from parents or guardians. If the need for consent was waived by the ethics committee, please include this information. All procedures involving human participants were approved by the North Carolina State University Institutional Review Board (IRB #26517). Informed consent was obtained from all participants before participation. Participants were informed of the voluntary nature of the study and provided digital consent before beginning the survey. The IRB approved a waiver of documentation of written consent due to minimal risk and the online administration of the survey. No minors were included. We have updated the text accordingly.

We note that the grant information you provided in the ‘Funding Information’ and ‘Financial Disclosure’ sections do not match. When you resubmit, please ensure that you provide the correct grant numbers for the awards you received for your study in the ‘Funding Information’ section. The Center for Precision Microbiome Engineering (PreMiEr) is a National Science Foundation (NSF)-funded Engineering Research Center (ERC) (Award # 2133504)

Thank you for stating the following financial disclosure:

“National Science Foundation”

Please include this amended Role of Funder statement in your cover letter; we will change the online submission form on your behalf. Thank you, we added this text to the disclaimer, “The funders had no role in study design, data collection and analysis, decision to publish, or preparation of the manuscript."

We note that you have indicated that there are restrictions to data sharing for this study. For studies involving human research participant data or other sensitive data, we encourage authors to share de-identified or anonymized data. However, when data cannot be publicly shared for ethical reasons, we allow authors to make their data sets available upon request. For information on unacceptable data access restrictions, please see http://journals.plos.org/plosone/s/data-availability#loc-unacceptable-data-access-restrictions.

a) If there are ethical or legal restrictions on sharing a de-identified data set, please explain them in detail (e.g., data contain potentially identifying or sensitive patient information, data are owned by a third-party organization, etc.) and who has imposed them (e.g., a Research Ethics Committee or Institutional Review Board, etc.). Please also provide contact information for a data access committee, ethics committee, or other institutional body to which data requests may be sent. The data underlying this study cannot be shared publicly due to ethical and legal restrictions imposed by the North Carolina State University Institutional Review Board (IRB). Although the dataset is de-identified, it contains potentially sensitive information related to participants’ health beliefs, religious practices, political orientations, and emotional responses to biotechnological interventions—data that, in combination, could pose a risk of deductive disclosure. The IRB determined that public release of the dataset, even in de-identified form, could not ensure adequate protection of participant privacy given the nature of the topics studied and the granularity of some variables. This determination aligns with established ethical standards in human subjects research that prioritize respect for persons, confidentiality, and the principle of minimizing harm. Access to the de-identified data may be requested on a case-by-case basis for qualified researchers who agree to strict confidentiality protocols and data use agreements. Requests should be directed to:

North Carolina State University Institutional Review Board

Email: irb-director@ncsu.edu

Phone: +1 (919) 515-4514

Website: https://research.ncsu.edu/compliance/irb/

Please include your full ethics statement in the ‘Methods’ section of your manuscript file. In your statement, please include the full name of the IRB or ethics committee who approved or waived your study, as well as whether or not you obtained informed written or verbal consent. If consent was waived for your study, please include this information in your statement as well. We have updated the text to read as follows, “This study was reviewed and approved by the North Carolina State University Institutional Review Board (IRB Protocol #26517). All participants provided informed consent prior to participation. Because the study posed minimal risk and was conducted online, the IRB approved a waiver of written documentation of consent. Participants provided digital consent after reading a disclosure statement describing the study’s purpose, procedures, and their rights as research participants. No personally identifiable information was collected, and all responses were stored and analyzed in de-identified form.”

We note that Figure 1 in your submission contain copyrighted images. All PLOS content is published under the Creative Commons Attribution License (CC BY 4.0), which means that the manuscript, images, and Supporting Information files will be freely available online, and any third party is permitted to access, download, copy, distribute, and use these materials in any way, even commercially, with proper attribution. For more information, see our copyright guidelines: http://journals.plos.org/plosone/s/licenses-and-copyright. We removed the copyrighted Microsoft Word icons and replaced them with Canva Free Content License icons that we edited in order to add our own creativity. You can read the Free Content License agreement here: https://www.canva.com/policies/content-license-agreement/. We ensured each of the three icons was covered by the Free Content License. Please see sections 2 and 6 for relevant information about the Free Content License.

Reviewer 1

The work by Cummings and collogues encapsulates an interesting investigation of the factors influencing one’s attitudes to using microbes as a preventative in preventing hospital acquired infections. There are many errors in the introduction where papers are cited that do not accurately support their claims. This reduces the reviewer’s enthusiasm for the work and their trust in the author’s presented results. We have updated a few of the references to better support the claims made in this paper.

- Personally, I would reduce the number of abbrevations. It makes the text overly verbose. Too many to keep track of. While we understand the concern, we have chosen to retain key abbreviations to preserve clarity and consistency with terminology common to microbiome engineering, infection control, and public health research. We have, however, reviewed the manuscript to ensure that all abbreviations are defined upon first use and that any non-essential abbreviations have been removed to minimize potential confusion.

- Reference to Rodrigues et al 2025 does not support using engineered bacteria, at no point in the cited study were probiotics used, however it was noted that women who were hospitalized were at a higher risk of acquiring an infection. We have updates this to Wright’s 2025 work: Wright, K. (2025, March 13). The Future of Microbiome‑Based Therapeutics. European Medical Journal, 10[1]:35-39, https://doi.org/10.33590/emj/CTGY2726

- Reference to Fady once again does not support the claim being made "For example, bioengineered waste lignocellulosic biochars have been shown to sequester MDR bacteria, reducing hospital contamination and minimizing antibiotic- resistant outbreaks (Fady et al., 2024).” We have updated the claim to appropriately reflect the Fady et al work. It now reads, “precision-engineered waste lignocellulosic biochars have been shown to sequester multidrug‑resistant clinical isolates in model wastewater systems (removing up to 94% of Pseudomonas aeruginosa and 85% of Staphylococcus aureus) — though their efficacy in clinical or hospital environments has not yet been evaluated”

- This paper is not about phages at all. Discusses nanoparticles to treat infections but does not mention either of the points it is meant to support “Additionally, phage therapy and microbiome restoration techniques have gained traction as alternative interventions, showing promise in targeting antimicrobial-resistant bacterial strains without relying on traditional antibiotics (Ahmed et al., 2025).” Phage therapy has existed as a concept since the early 1900s, there should be at least a few references which would be better used to support this claim. We have deleted this reference and adjusted the text.

- Another instance of an improper citation: “Recurrent bacteremia in the setting of Pseudomonas endocarditis of the tricuspid valve and indwelling medical devices”, cited work is case report, regarding biofilm formation on a medical device. No review of supplying probiotics to outcompete pathogens as suggested by the quoted line. Thank you, we have corrected the text and added another citation that better supports the claims as follows, “Although recurrent infections associated with Pseudomonas and biofilm formation on medical devices remain a serious clinical challenge (Miller et al., 2024), emerging studies suggest that introduced non-pathogenic microbes may help outcompete pathogenic strains in certain contexts. For example, experimental trials using engineered bacterial consortia have demonstrated success in displacing Klebsiella pneumoniae and other multidrug-resistant pathogens in controlled environments (Sheth et al., 2022).”

Sheth, R. U., Cabral, V., Chen, S. P., & Wang, H. H. (2022). Manipulating bacterial communities by in situ microbiome engineering. Nature Reviews Microbiology, 20(5), 365–378. https://doi.org/10.1038/s41579-021-00641-1

- There is no Cummings 2025 in the works cited. I assume that this is in reference to the manuscript itself, which seems odd if not misleading: “So far there have been limited studies of U.S. public attitudes on microbiome engineering in the built environment. In the first survey of built microbiome engineering and public attitudes, attitudes toward microbes and microbiome engineering were discovered to be shaped by a complex interplay of scientific literacy, emotional responses, media influence, and demographic factors (Cummings, Landreville, & Kuzma, 2024; Cummings, Landreville, & Kuzma, 2025)”. Thank you for pointing this out—it is a reference to our team’s recent work and we have added the proper citation to the reference list as follows,

Cummings CL, Landreville KD and Kuzma J

(2025) Natural vs. genetically engineered

microbiomes: understanding public attitudes

for indoor applications and pathways for

future engagement.

Front. Genet. 16:1560601.

doi: 10.3389/fgene.2025.1560601

- It would be interesting to see if one’s prior scientific knowledge was in anyway predictive of their thoughts on the microbiome in therapy, perhaps including this information in the model would provide interesting insights as to whether education is actually helpful in increasing the public’s perception of microbiome based therapeutics While this is a compelling area for research, it falls outside the scope of the current study. Interestingly, prior research has consistently shown that increasing scientific knowledge alone—sometimes referred to as the “scientiation” approach—does not reliably lead to more positive attitudes toward emerging biotechnologies. Studies in science communication and public understanding of science have demonstrated that emotional responses, trust in institutions, and cultural values often play a more significant role in shaping public perceptions than factual knowledge alone (e.g., Cummings et al., 2024; Kahan, 2015; Allum et al., 2008). Nonetheless, we agree that understanding how specific types of knowledge may interact with other attitudinal drivers would be a valuable direction for future research.

Reviewer 2

Cummings and colleagues present a very thorough, well written, and interesting assessment of Americans’ attitudes towards introducing engineered microbiomes into hospital sinks to combat antimicrobial resistance and hospital associated infections. The study is timely as AMR and HAI are major issues facing the US healthcare system and innovative strategies are needed to combat them. Additionally, deploying engineered microbiomes in the built environment is in early stages, making now an ideal time for assessing public attitudes so that they may inform future implementation strategies and promote success. I think the manuscript can be improved by adding some detail/clarifying information, tightening up some of the language to describe survey respondent characteristics, and including a discussion of the study’s limitations. My specific suggestions are below. Thank you for the positive feedback!

Major comments

1. I think it would be nice to give a little detail/context on why the study/survey focused on introduced microbiomes specifically deployed in sinks (as opposed to other parts of the hospital built environment). We fully agree and have added clarifying language in the manuscript to explain the focus on hospital sinks as the deployment site for introduced microbiomes. It know reads as follows, “This study focuses specifically on hospital sinks as the target site for introduced microbiomes due to their well-documented role as persistent reservoirs for multidrug-resistant organisms such as Pseudomonas aeruginosa, Klebsiella pneumoniae, and Acinetobacter baumannii. These pathogens frequently colonize sink drains and plumbing systems, where they can form biofilms, exchange resistance genes, and contribute to repeated contamination events within clinical environments. Sinks represent one of the few built-environment surfaces in hospitals currently being explored for targeted microbial interventions, including probiotic and synthetic consortia-based approaches. As such, they offer a realistic and policy-relevant use case for gauging public support for microbiome engineering in infection control.”

2. I think the last sentence of the introduction – very clearly stating the overall research question – is really great and sets the reader up nicely for the rest of the manuscript! (no revisions requested here) Thanks!

3. Are there any notable examples/success stories of deploying engineered/introduced microbiomes in other built environment settings (healthcare or otherwise)? If yes, I would consider giving a brief overview of these in the literature review section, along with any lessons learned that may be relevant to the current study. Alternatively, are there any notable failures with relevant lessons learned? We appreciate the reviewer’s thoughtful suggestion to include examples of successful or unsuccessful deployments of engineered or introduced microbiomes in built environments. While this is an important area of future inquiry, the field remains nascent, with fe

---

## [Decision Letter · Decision Letter 1]

3 Sep 2025

Public Perceptions and Support for Introduced Microbes to Combat Hospital-Acquired Infections and Antimicrobial Resistance

PONE-D-25-30050R1

Dear Dr. Cummings,

We’re pleased to inform you that your manuscript has been judged scientifically suitable for publication and will be formally accepted for publication once it meets all outstanding technical requirements.

Kind regards,

Assoc. Prof. Phakkharawat Sittiprapaporn, Ph.D.

Academic Editor

PLOS ONE

Additional Editor Comments (optional):

Reviewer #2:

Reviewer #3:

Reviewers' comments:

Reviewer's Responses to Questions

**Comments to the Author**

1. If the authors have adequately addressed your comments raised in a previous round of review and you feel that this manuscript is now acceptable for publication, you may indicate that here to bypass the “Comments to the Author” section, enter your conflict of interest statement in the “Confidential to Editor” section, and submit your "Accept" recommendation.

Reviewer #2: All comments have been addressed

Reviewer #3: All comments have been addressed

2. Is the manuscript technically sound, and do the data support the conclusions?

Reviewer #2: Yes

Reviewer #3: Yes

3. Has the statistical analysis been performed appropriately and rigorously? 

Reviewer #2: Yes

Reviewer #3: Yes

4. Have the authors made all data underlying the findings in their manuscript fully available?

Reviewer #2: No

Reviewer #3: Yes

5. Is the manuscript presented in an intelligible fashion and written in standard English?

Reviewer #2: Yes

Reviewer #3: Yes

6. Review Comments to the Author

Reviewer #2: Thank you for your thoughtful and thorough responses and revisions! I have no further comments.

Signed: Kayla A. Carter

Reviewer #3: The authors answered all of my questions thoughtfully and thoroughly. The provided easy-to-understand explanations about the statistical analysis. This helped me understand their approach. The manuscript reads very well.

7. PLOS authors have the option to publish the peer review history of their article (what does this mean? ). If published, this will include your full peer review and any attached files.

**Do you want your identity to be public for this peer review?** For information about this choice, including consent withdrawal, please see our Privacy Policy .

Reviewer #2: **Yes: ** Kayla A. Carter

Reviewer #3: No

---

## [Editor Report · Acceptance letter]

PONE-D-25-30050R1

PLOS ONE

Dear Dr. Cummings,

I'm pleased to inform you that your manuscript has been deemed suitable for publication in PLOS ONE. Congratulations! Your manuscript is now being handed over to our production team.

Kind regards,

on behalf of

Assoc. Prof. Dr. Phakkharawat Sittiprapaporn

Academic Editor

PLOS ONE